# Spatial and Temporal Dynamics of the Vegetation Cover from the Bijagual Massif, Boyacá, Colombia, during the 1986–2021 Period

**DOI:** 10.3390/plants13070948

**Published:** 2024-03-25

**Authors:** Pablo Andrés Gil-Leguizamón, Jaime Francisco Pereña-Ortiz, Daniel Sánchez-Mata, Ángel Enrique Salvo-Tierra, Jorge David Mercado-Gómez, María Eugenia Morales-Puentes

**Affiliations:** 1Doctoral Program in Biological and Environmental Sciences, SisBio, Herbario UPTC, ViE-DIN, Universidad Pedagógica y Tecnológica de Colombia, Campus Universitario, Tunja 150003, Colombia; pablo.gil@uptc.edu.co (P.A.G.-L.); maria.morales@uptc.edu.co (M.E.M.-P.); 2Department of Botany and Plant Physiology, Faculty of Sciences, Universidad de Málaga, 29010 Málaga, Spain; jportiz100@hotmail.com (J.F.P.-O.);; 3Botany Unit, Department of Pharmacology, Pharmacognosy and Botany, Faculty of Pharmacy, Complutense University, 28040 Madrid, Spain; 4Harvard University Herbaria, Department of Organismic and Evolutionary Biology, Harvard University, Cambridge, MA 02138-2094, USA; 5Grupo Evolución y Sistemática Tropical, Department of Biology and Chemistry, Universidad de Sucre, Sincelejo 700001, Colombia; jorge.mercado@unisucre.edu.co

**Keywords:** high Andean forest, Landsat, landscape metrics, land use change, multitemporality, paramo

## Abstract

Landscape changes based on spectral responses allow showing plant cover changes through diversity, composition, and ecological connectivity. The spatial and temporal vegetation dynamics of the Bijagual Massif from 1986 to 2021 were analyzed as a measure of ecological integrity, conservation, and territory. The covers identified were high open forest (Hof), dense grassland of non-wooded mainland (Dgnm), a mosaic of pastures and crops (Mpc), lagoons (Lag), and bare and degraded lands (Bdl). The Bijagual Massif has 8574.1 ha. In 1986, Dgnm occupied 42.6% of the total area, followed by Mpc (32.8%) and Hof (24.5%); by 2000, Mpc and Hof increased (43.7 and 28.1%, respectively), while Dgnm decreased (28%); by 2021, Dgnm was restricted to the northeastern zone and continued to decrease (25.2%), Mpc occupied 52.9%, Hof 21.7% and Bdl 0.1%. Of the three fractions of the connectivity probability index, only *dPCintra* and *dPCflux* contribute to ecological connectivity. Hof and Dgnm show patches with biota habitat quality and availability. Between 1986 and 2021, Dgnm lost 1489 ha (41%) and Hof 239.5 ha (11%). Mpc replaced various covers (1722.2 ha; 38%) in 2021. Bijagual has a valuable biodiversity potential limited by Mpc. Territorial planning and sustainable agroecological and ecotourism proposals are required due to the context of the ecosystems.

## 1. Introduction

High mountain ecosystems in Colombia are characterized by a degree of endemicity and richness of taxonomic entities [1,2] due to the outcome of landscape and topographic evolution, as well as climatic modeling and ecological interaction [3,4]. For example, plant physiognomy, functionality, and diversity have allowed the identification of ecological associations related to different altitudinal belts, such as high Andean forest and paramo [5,6,7,8]. In fact, these ecological associations have been considered strategic [9] because they provide ecosystem services such as food and climate, air and soil regulation, and water capacity filtration and concentration [10,11]. However, for example, in the Andean forest, human transformations have shaped the native forest landscapes into a matrix of forest patches, cultivated and grazing areas, and socio-ecosystem interactions or relationships [12,13]. To these directly caused human transformations, other indirectly caused ones due to the current situation of global climate change are added, mainly, the constant increase in the average annual temperature and the decrease in precipitation over the last decades. These climatic anomalies cause, among other effects, alterations in the natural dynamics of vegetation, changes in the structure and composition of their communities, and a notable loss of biodiversity [14,15,16]. In the case of paramo formations, given their high sensitivity to environmental disturbances, it is estimated that more than three-quarters of the original ecosystem has been altered due to the increase in temperatures, which dries the soil and forces species to move along the altitudinal axis towards areas more suitable for their development [17,18].

Due to the socio-ecosystem relationship that occurred in this region since the 19th century [19,20,21], the vegetation cover changes over time can be identified and, hence, produce accurate information on the distribution and rates of vegetation cover chances, particularly regarding forest and paramo, which are essential for conservation strategies [22,23]. Change detection at the landscape scale can be identified by analyzing the spatiotemporal effect of vegetation reflectance [24,25,26]. Spectral variations model the vegetation reflectance using combinations of satellite image bands [22,26]. Furthermore, because Landsat satellite images can be obtained from 1970, the historical changes in plant covers can be identified over time [25,26].

In Colombia, some Andean forest and paramo areas are delimited by administrative units called complexes [27], which include the Bijagual Massif under the Tota-Bijagual-Mamapacha complex. The land use and forest deforestation history of Bijagual has been related to intensive and extensive agricultural and livestock activities [28,29,30]. However, the magnitude of human activities under the spatiotemporal effect of vegetation reflectance and, therefore, the historical, spatial, and temporal dynamics of fragmentation and ecological connectivity are unknown [31,32].

Accordingly, this work studied the spatial and temporal dynamics of the Bijagual Massif vegetation in Colombia to measure the ecological integrity of the natural area replaced by excessive human activities [30]. The aim of the above analysis is to provide reference values for the conservation, ecology, sustainability, vegetation management, and territorial planning of the last remnants of forest and paramo existing in the Bijagual Massif, which is considered an ecological corridor, regulating high-mountain water and climate in Colombia [4,5,9,11,27,30]. Based on remote sensing data, we explore the supervised classification method in two periods spanning 35 years, from 1986–2000 and 2000–2021. This study is based on: (1) Describing the natural high-mountain vegetation covers in Bijagual as an indicator of their potential biodiversity. (2) Validating the thematic reliability (spectral separability) of the current vegetation covers. (3) Identifying spatiotemporal changes (losses and gains) in high-mountain natural areas and their influence on ecological connectivity. The results and their analyses are explained in three sections: (1) Land covers—field verification, (2) Classification accuracy, and (3) Multitemporal dimension.

## 2. Results

### 2.1. Land Covers—Field Verification

Field verification allowed the identification of five cover types spectrally assigned to the Corine Land Cover categories (CLC; IDEAM [33]), as follows: 1. High open forest (Hof): includes the high Andean forest formation dominated by trees > 10 m tall; 2. Dense grasslands of non-wooded mainland (Dgnm): belongs to the paramo ecosystem characterized by scrubs, pajonal-chuscal, frailejonal, and sphagnal type communities; 3. Mosaic of pastures and crops (Mpc): lands whose use-occupation are pastures and crops, particularly potatoes; 4. Lagoons, lakes, and natural swamps (Lag): natural water surfaces that, for Bijagual, correspond to the La Calderona and El Pato lagoons; 5. Bare and degraded lands (Bdl): deforested area (Figure 1).

The vegetation of the Bijagual Massif belongs to Andean mountain formations that characterize the high mountain ecosystems in Colombia. According to Gil-Leguizamón et al. [34,35], the strip of high Andean forest (Hof) shows a wide species turnover at short distances, with trees and shrubs of encenillos (*Weinmannia tomentosa* L. f., *W. rollottii* Killip, *W. fagaroides* Kunth, *W. balbisiana* Kunth, and *W. reticulata* Ruiz & Pav.), gaque (*Clusia alata* Planch. & Triana, *C. elliptica* Kunth, and *C. multiflora* Kunth), susque (*Brunellia propinqua* Kunth, and *B.* cf. *comocladifolia* Bonpl.), garrocho (*Viburnum tinoides* L. f., and *V. triphyllum* Benth.), uva camarona (*Macleania rupestris* (Kunth) A.C. Sm.), tunos, charne, and siete cueros (*Miconia cataractae* Triana, *M. cundinamarcensis* Wurdack, *M. jahnii* Pittier, *M. ligustrina* (Sm.) Triana, *M. theizans* (Bonpl.) Cogn., *Axinaea scutigera* Triana, *Bucquetia glutinosa* (L. f.) DC., and *Tibouchina grossa* (L. f.) Cogn.), and manos de oso (*Oreopanax bogotensis* Cuatrec., and *O. mutisianus* (Kunth) Decne. & Planch.), among other species identified in field verification visits and corroborated with the collections of the UPTC Herbarium of Universidad Pedagógica y Tecnológica de Colombia.

In the dense grassland of non-wooded mainland or paramo coverage (Dgnm), there are associations of scrubs, pajonales-chuscales, frailejonales, and sphagnals included in the *Linochilo phylicoidis-Arcytophyllion nitidi* Rangel & Ariza 2000 *nom. mut. nov. (all.)* alliance described by Pinto Zárate [36] for the Eastern Cordillera of Colombia. Shrub communities include species such as *Bejaria resinosa* Mutis ex. L. f., *Escallonia myrtilloides* L. f., *Miconia cleefii* L. Uribe, *Geissanthus andinus* Mez, *Ageratina theaefolia* (Benth.) R.M. King & H. Rob., *Diplostephium phylicoides* (Kunth) Wedd., *Pentacalia pulchella* (Kunth) Cuatrec., *Monnina salicifolia* Ruiz & Pav., *Hesperomeles obtusifolia* (Pers.) Lindl., and *Symplocos theiformis* (L. f.) Oken., among others. The pajonal-chuscal formations are comprised of *Agrostis perennans* (Walter) Tuck., *Anthoxanthum odoratum* L., *Calamagrostis effusa* (Kunth) Steud., and *Chusquea tessellata* Munro., among others. Frailejonal formations include *Espeletia argentea* Bonpl., *E. boyacensis* Cuatrec., and *E. murilloi* Cuatrec., and the sphagnals formation comprises a complex of *Sphagnum* species [34,37].

Endemic and endangered species are found in the Bijagual Massif, including forest formation species, such as *Greigia stenolepis* L.B. Sm., *Hieronyma rufa* P. Franco, *Tillandsia pallescens* Betancur & García Nestor, *Symplocos venulosa* Cuatrec., *Diplostephium oblongifolium* Cuatrec., and *Dunalia trianaei* Dammer [30]. Furthermore, *Espeletia cayetana* (Cuatrec.) Cuatrec., *E. murilloi* Cuatrec., and *Puya goudotiana* Mez. [37] were found in the paramo ecosystem. In addition, species widely distributed in the high Andean forests and paramos in Boyacá were recorded, including angelitos (*Monochaetum myrtoideum* Naudin, *M. uribei* Wurdack, and *M. meridense* Naudin), chites (*Hypericum laricifolium* Juss., *H. mexicanum* L., *H. myricariifolium* Hieron., and *H. lycopodioides* Triana & Planch.), and cortadera (*Cortaderia nitida* (Kunth) Pilg.) [38].

The species composition in the Bijagual Massif is related to a high biodiversity potential [34]. Nevertheless, vegetation is limited by the pastures and crops cover (Mpc) mosaics extending from south to north in the study area. This cover type is dominated by potato (*Solanum tuberosum* L.), mora (*Rubus urticifolius* Poir.), and beans (*Phaseolus vulgaris* L.), as well as grazing cattle and sheep, considered potential threats to the ecological integrity of the verified ecosystems [39].

### 2.2. Classification Accuracy

During the field verification, 276 control points were generated. Producer and user accuracy fluctuated between 98 and 100% of the covers. The Kappa coefficient established a thematic reliability of 95% (gray diagonal boxes; Table 1).

### 2.3. Multitemporal Dimension

The Bijagual Massif has a total landscape area (TA) of 8574.1 ha. In 1986, Dgnm (paramo) occupied 42.6% of the total area, followed by Mpc (32.8%) and Hof (24.5%). In 2000, the spatial composition was maintained; however, the productive systems and the forest increased (43.7% and 28.1%, respectively), while the paramo decreased (28%). In 2021, Bijagual showed paramo extensions restricted to the northeast in the municipality of Viracacha and isolated patches in the center and south (2160 ha; 25.2% of the total area). Mpc occupies 52.9% (4534.1 ha) and Hof 21.7% (1861.8 ha; Table 2; Figure 1).

Between 1986 and 2000, Dgnm lost 1248 ha (34% of the total area), while Mpc and Hof increased 936.6 ha (25% of the total area) and 309.2 ha (13% of the total area), respectively. From 2000 to 2021, Dgnm and Hof lost area (241 ha, equivalent to 10% and 548.7 ha or 23%, respectively). Furthermore, Mpc increased (785.6 ha or 17%). In 35 years, paramo and forest areas decreased (1489 ha or 41% and 239.5 ha or 11%, respectively), while pastures and crops increased (1722.2 ha or 38%) and occupied more than 50% of the land extension of the Bijagual Massif (Table 2; Figure 1).

#### 2.3.1. Space-Time Changes in High Open Forest (Hof) Coverage

From 2101.3 ha of Hof registered in the year 1986, 1875.8 ha (89%) were conserved by the year 2000. In this same year, Hof increased by 534.8 ha, which were obtained from the coverages Dgnm (342.5 ha), Mpc (191.4 ha), and Lag (0.9 ha); the total area of Hof in the year 2000 was 2410 ha. By the year 2021, Hof preserved 1441.3 ha (60% of the total estimated in the year 2000) and increased by 420.5 ha obtained from Dgnm (257.9 ha), Mpc (161.6 ha), and Lag (1.0 ha); the total area of Hof in the year 2021 was 1861.8 ha (Figure 2). Overall, an increase in area was evident for the period 1986–2000, followed by a decrease for the period 2000–2021. The area of Hof for the year 2021 was lower than the area estimated for the year 1986 (Table 2, Figure 2).

#### 2.3.2. Space-Time Changes in Dense Grassland of Non-Wooded Mainland (Dgnm) Coverage

From 3648.9 ha of Dgnm registered in the year 1986, only 2123.6 ha (58%) were conserved by the year 2000. In this same year, Dgnm increased by 277.4 ha, obtained from Hof (130.6 ha) and Mpc (146.8 ha); the total area of Dgnm in the year 2000 was 2401 ha. By the year 2021, Dgnm preserved 1402.5 ha (58% of the total estimated in the year 2000) and increased by 757.6 ha obtained from Hof (447.3 ha) and Mpc (310.3 ha); the total area of Dgnm in the year 2021 was 2160 ha (Figure 2). Overall, the area of Dgnm for the year 2021 was lower than the area estimated for the year 1986 (Table 2, Figure 2).

#### 2.3.3. Space-Time Changes in Mosaic of Pastures and Crops (Mpc) Coverage

From 2811.8 ha of Mpc registered in the year 1986, only 2473.6 ha (88%) were conserved by the year 2000. In this same year, Mpc increased by 1274.9 ha obtained from Dgnm (1182.8 ha) and Hof (92.1 ha); the total area of Mpc in the year 2000 was 3748.5 ha. By the year 2021, Mpc preserved 3271.6 ha (87% of the total estimated in the year 2000) and increased by 1262.5 ha obtained from Dgnm (740.6 ha) and Hof (521.9 ha); the total area of Mpc in the year 2021 was 4534.1 ha (Figure 2). Overall, the area of Mpc for the year 2021 was greater than the area estimated for the year 1986 (Table 2, Figure 2).

Between 1986 and 2000, of the total area of 8574.1 ha, 6484.3 ha maintained the same coverage (75.6%), and 2089.9 ha changed to another cover type (24.4%). Between 2000 and 2021, only 6128.6 ha maintained its cover (71.5%), and 2445.5 ha changed from one to another (28.5%) (Figure 3).

#### 2.3.4. Diversity

In 1986, 76 patches (TNP) corresponded to natural or transformed areas, with a density (PD) of 0.88 patches/ha. In 2000, 75 patches (PD: 0.87 patches/ha) were recorded, and in 2021, 396 patches (PD: 4.6 patches/ha) were found.

#### 2.3.5. Composition

From the total number of patches in 1986, 25 belong to Mpc (PD: 0.3 patches/ha), but these decreased in 2000 (NP: 21 patches & PD: 0.24 patches/ha) and increased by 2021 (NP: 74 & PD: 0.86 patches/ha). For Mpc in 1986, the largest patch index (LPI) was 18.4%, rising to 22.7% in 2000 and 48.7% in 2021. In Dgnm, the number of patches between 1986 and 2000 increased (from NP: 25 & PD: 0.3 patches/ha to NP: 31 & PD: 0.36 patches/ha); however, the number of patches was higher in 2021 (NP: 168 & PD: 1.9 patches/ha). Conversely, the LPI values of this coverage decreased in 35 years (1986: 21%; 2000: 8.1%; 2021: 6.7%). The abundance of Hof patches decreased between 1986 and 2000 (from NP: 23 & PD: 0.25 patches/ha to NP: 20 & PD: 0.23 patches/ha) but increased in 2021 (NP: 150 & PD: 1.7 patches/ha). The LPI index for Hof shows a decrease over time (1986: 8.5%; 2000: 8.2%; 2021: 6.5%; Figure 4).

#### 2.3.6. Connectivity

Of the three fractions of the *dPCk* probability index, only *dPCintra* and *dPCflux* contribute to connectivity, while the contribution of the *dPCconnector* fraction is limited (Figure 5A).

In 1986 and 2000, the Hof and Dgnm covers showed patches with habitat quality and availability for biota (connectivity within the patch) and act as dispersal flows (connectivity between patches); however, these ecological characteristics decreased drastically in 2021 for the two covers. Both Hof and Dgnm patches provide different connectivity values, evidenced in the dispersion of the standard deviation data concerning the average of each fraction by cover type and time (Figure 5A).

Habitat availability contributions and flow between patches in 1986 and 2021 were low (*dPCintra* < 10 and *dPCflux* < 5; Figure 5B). In the spatiotemporal dynamics, changes in the connectivity contributions of each patch were identified for the *dPCintra* and *dPCflux* fractions. In this way, in 1986, Hof showed two patches that contribute to habitat quality and availability. In 2000, one patch contributed to habitat quality and availability, two contributed to habitat quality and ecological flow, and the other two to ecological flow (Figure 5B). In 2021, only one patch contributed to habitat quality and availability and five to ecological flow (Figure 5B,C). For its part, Dgnm in 1986 showed one patch with contributions to habitat quality, habitat availability, and ecological flow (or dispersion), another with contributions to habitat quality and availability, and solely one patch with contributions to ecological flow. In 2000, only one patch registered contributions to habitat quality and availability (Figure 5B), and in 2021, one patch contributed to habitat quality, habitat availability, and ecological flow, and another solely to ecological flow (Figure 5B,C).

## 3. Discussion

### 3.1. Land Cover—Field Verification

The Bijagual Massif corresponds to one of the six high mountain natural areas in Boyacá with broad floristic diversity [27,38]. The field verification process and the consultation of herbarium biological collections [35,37] allowed estimating a richness of up to 327 species for the Hof cover and up to 120 species for Dgnm. The results of the spatiotemporal dynamics showed that this diversity is vulnerable to the effects of the pasture and crop matrix since it has caused modifications in the vegetation [40]. According to Gil-Leguizamón et al. [34,35], the presence and dominance of some species of *Chusquea* Kunth in forest interiors and of *Cenchrus* L., *Brachiaria* (Trin.) Griseb., and *Holcus lanatus* L. in paramos indicate that the composition and physiognomy of these covers have been modified, enhancing the loss of richness and forming more heterogeneous plant communities. Even so, the research carried out by Gil-Leguizamón et al. [34,35] and Carrillo et al. [37] relate some species of ecological importance, including trees, shrubs, rosettes, and grasses that characterize the physiognomy of the Andean forest, such as *C. multiflora*, *C. elliptica*, *C. alata*, *W. rollottii*, *B. comocladifolia*, *V. triphyllum*, and *E. myrtilloides*, and then the species *Espeletia murilloi* Cuatrec., *Paepalanthus columbiensis* Ruhland, *Puya goudotiana* Mez, *Hypericum lycopodioides* Triana & Planch., *Blechnum auratum* (Fée) R. M. Tryon & Stolze, *Calamagrostis effusa* (Kunth) Steud., and *Rhynchospora ruiziana* Boeckeler typical of paramos. These records agree with the altitudinal delimitation of the vegetation proposed by Cuatrecasas [41].

Research is required for these groups to allow inferring biogeographic relationships, particularly of the Andean forest, which, through distribution records, are not only indicators that corroborate land cover but also allow interpreting the influence of the environmental gradients in the distribution patterns of the flora, its endemisms, and affinities with other biogeographic units, as has been done for the non-vascular flora of Bijagual [42].

Besides generating thematic reliability in land cover layers, field verifications allow identifying species that, due to their phenological, reproductive, and dispersal characteristics, can become a potential for ecological restoration processes in Bijagaul [39,43]. Within these, *Myrcianthes rhopaloides* (Kunth) McVaugh, *Pentacalia pulchella* (Kunth) Cuatrec., *Diplostephium floribundum* (Benth.) Wedd., *Vallea stipularis* L. f., *Weinmannia fagaroides* Kunth, *Macleania rupestris* (Kunth) A.C. Sm., *Hypericum lycopodioides* Triana & Planch., *Bucquetia glutinosa* (L. f.) DC and some species of *Viburnum* L. and *Cestrum* L. are highlighted.

### 3.2. Classification Accuracy

From the 276 control points obtained during the field verification and the overlap of these points with the land cover raster layer (spectral response obtained from the supervised classification for the 2021 image), a thematic reliability of 95% (Kappa 0.95) was obtained. According to cartographic quality standards, the data are associated with a “very good spatial concordance” [44]. Of these 276 points, nine did not coincide with their respective classification cover due to the similarity of the digital value of a pixel in the image associated with another cover. This was evidenced for Hof with six points corresponding to forest cover in the field verification and that in the supervised classification algorithm were assigned to Dgnm (4 points) and Mpc (2 points). A similar case was obtained for Lag with only two points and Dgnm with one (Table 1). The remaining 267 control points maintain a concordance between verification and classification, which allowed producer and user accuracy to be higher than 98%, and the errors of omission and commission were not higher than 2% (Table 1). For this reason, the pre- and post-processing images were considered reliable.

### 3.3. Multitemporal Dimension

Changes in plant cover due to changes in land use affect the properties and functioning of ecosystems [45,46]. The vegetation and natural covers in the Colombian Andean relief have been modified as a result of human colonization processes. For 35 years in Bijagual, extensive and small areas of forest and paramo were substituted for agricultural and livestock activities, which currently affect habitat loss for biodiversity [28,47]. The landscape structure [48], species dynamics [5], and the ecological network conformation of the natural space of the Tota-Bijagual-Mamapacha complex [27], currently threatened by human activities, have caused a decrease in forest area and an increase in the natural distance of patches, and, therefore, the loss of this ecological continuity corridor [49,50,51] (Figure 1; Table 2).

Forest fragmentation has been more evident since 2021 due to the increase of Dgnm and Hof in the number of patches, but also due to area reduction and patch isolation. For 35 years, 1923.4 and 614 ha of Dgnm and Hof, respectively, were converted to Mpc (Figure 2), supported by the increase in the number of patches, the area, and the ones that become connected (Figure 1 and Figure 4).

Areas around the massif, which have been proposed as transition zones, were found to be affected by crops and livestock activities long before the 1980s. According to Rodríguez-Eraso et al. [52], the loss of natural area in the 80s and 90s was centered in the eastern Cordillera in Boyacá and Cundinamarca, caused by deforestation patterns of the high Andean forest and paramo due to food demand. In addition, Sánchez-Cuervo et al. [29], in the departments of Santander and Boyacá, pointed out that potato cultivation increased from 380 to 482 km^2^ between 2006 and 2008, reducing forest covers. Potato cultivation has a high per capita consumption in Colombia, and, therefore, expansion has occurred under native forest areas, increasing the continuous degradation of Andean forests and the paramos (Figure 2 and Figure 3).

Furthermore, another factor influencing forest cover is climate change, which has been enhancing during the last few years and has affected species distribution [53,54]. Since the Andean high mountain ecosystems are seasonal, the increase in temperature would prolong the intensity of the dry periods, and the paramo would reduce the capacity to capture and regulate water [40]. In this regard, the negative effects of human activities on natural ecosystems are exacerbated by the effects of climate change [55], particularly those that exhibit greater fragility due to their location, requirements, and functioning, such as paramos or high Andean forests, where the capacity for adaptation or colonization of new territories is very limited [56]. Applying the models of Yates et al. [57] and Lugo et al. [58], based on simple correlations between climatic variables and vegetation, in the Colombian departments of Casanare, Meta, Santander, La Guajira, as well as in the western sector of the departments of Chocó, Valle del Cauca, and Nariño, Alarcón and Pabón [59] detected a clear trend towards an increase in territorial occupation and coverage of vegetation formations that prefer drier conditions and lower altitudes, unlike those that require higher humidity and elevation conditions, such as these paramos and high Andean forests, which would be displaced from their current distribution.

This effect of substitution of some formations for others or changes in their coverage, related to climatic variations, has also been observed by other authors in different parts of the planet, such as Mendoza et al. [60] on the Pacific slope of Nicaragua, where areas of dry or very dry occupation have increased; Villers-Ruiz and Trejo-Vázquez [61], who highlighted how temperature increases in Mexico could favor the establishment of tropical vegetation at the expense of losing temperate and semi-cold forests of hardwoods and conifers; or Esquivel et al. [55] on the summits of Teide volcano (Canary Islands-Spain), where they observed the difficulty of some species and plant communities adapted to cool and humid environments to thrive in territories where there has been a significant rise in temperatures.

The paramo cover (Dgnm) decreased in area between 1986 and 2021 (1489 ha), while the forest expanded its area between 1986 and 2000 (309.2 ha) and decreased between 2000 and 2021 (548.7 ha). The forest area that decreased in the second period is larger than the area that expanded in the first (with a difference of 239.5 ha) (Table 2). Instead, the pasture and crop cover increased by 1722.2 ha in 35 years. Therefore, it is necessary to conserve the Dgnm and Hof areas that still exist in the municipalities of Viracachá, Cienaga, and Tibana (4034.9 ha [20,23,51]) and include structure, functionality, and ecological diversity attributes in the Bijagual extension [6,39,43], particularly in the municipality of Ramiriquí, which is the most affected by the expansion of the agricultural and livestock frontier (Figure 1). In the Colombian eastern mountain range, only 36.4% of the total paramo area (322.925 ha) and 25.3% of the Andean montane forest (427.675 ha that include the sub-Andean and Andean forest belts) are protected. However, of these areas, only 41 and 45% maintain their original extension [50]. The remaining areas have experienced land use changes with subsequent biodiversity loss [40].

Spectral image response outcomes show that natural succession has occurred around the paramo and forest areas (Figure 1 and Figure 2) [28,62,63]. Nevertheless, this process seems to be slow. Lequerica et al. [64] and Prado-Castillo et al. [39] suggest that succession is slow in areas previously used for cultivation due to degradation, and the compaction of soil produced by agriculture can limit the development of propagules and modifies the germinable or viable seed bank [65]. Hence, natural succession needs to be assisted by ecological restoration processes to improve the increase of paramo vegetation.

The increase in the total number of patches (TNP 76 to 396 between 1986 and 2021), density (PD), and percentage of patch occupation (LPI) are evidence of the intensive and extensive land use and the changes from one coverage to another. While in 1986, one patch was registered for each hectare, in 2021, four or five patches (PD) were recorded for each hectare. In 2021, Mpc coverage presented the lowest number of patches compared to Dgnm and Hof (NP 74, 168, and 150, respectively; Figure 4). However, Mpc presents patches with a larger occupation area compared to Dgnm and Hof (LPI 48.7, 6.7, and 6.5%, respectively; Figure 4). In 35 years, the LPI metric describes greater spatiotemporal occupancy of the Mpc patches and lower occupancy of the Dgnm and Hof patches (Figure 4).

The cover and topological position of the Bijagual landscape per patch allow the identification of different contributions to ecological connectivity. Each patch can act as a space in which there is connectivity, i.e., a larger area with a suitable habitat condition or quality that may have higher connectivity [66,67]. The functionality of some patches is evident, even more over those that show higher habitat availability (Figure 5B,C) since these can act as refuge, feeding and reproduction areas or are habitat units that produce or receive dispersal flows of individuals to other units or patches [67]. In fact, these areas can be analyzed in the metapopulation dynamic to establish possible unknown connections with the biotic complex of Tota to the north and Mamapacha to the south, which, together, comprise a biological corridor.

The spatial configuration of the Hof and Dgnm patches has contributed to the ecological connectivity in Bijagual from 1986 to 2021, evidenced by the contributions of the *dPCintra* and *dPCflux* fractions; however, during this period, also natural area loss has occurred, affecting the ecological diversity [34,68,69]. Ecological connectivity in Bijagual decreased noticeably from 2000 to 2021 (Figure 5A). The massif may experience a fragmentation process in the near future [20,49,50], and the current landscape of this natural area shows a loss in habitat quality and availability [49,66,67].

As a result of the distancing and isolation of the Dgnm and Hof patches, the continuity of the ecological corridor has been interrupted (Figure 1). This, in turn, poses difficulties for the movement of species and genetic exchange [67,70] and is evident in the reduced contribution of the *dPCconnector* fraction (Figure 5A) that determines limited dispersion fluxes. However, the Bijagual Massif still has paramo and Andean forest patches that can act as critical habitat areas for maintaining ecological connectivity (Figure 5C), where habitat loss or deterioration would have an even more negative impact on landscape connectivity. These areas must be prioritized and considered reference ecosystems for ecological restoration processes.

This research states that the Bijagual Massif requires the inclusion of sustainable management practices and territorial planning with agroecological, forestry, environmental education, ecotourism, biogeographical, and climate research proposals, according to the natural environment considering the strategic and vulnerable ecosystem context where the State interacts through its territorial entities (government and environment), the peasant communities (through citizen participation actions of the associated communities), agricultural companies, and the academy as knowledge generators.

## 4. Materials and Methods

### 4.1. Study Area

The Bijagual Massif is located in the Andes eastern mountain range over the Department of Boyacá (Colombia) between 5°26′0.0″–5°16′0.0″ N and 73°14′0.0″–73°22′0.0″ W [34]. The vegetation is composed of paramo (between 2990 and 3460 m a.s.l.) and high Andean forest (between 2682 and 3268 m a.s.l.) in an extension of 8574.1 ha (Figure 6). Its borders delimit to the north with Lake Tota and the paramos of Toquilla, Sarna, Suse, and Los Curíes and to the south with the Mamapacha Massif. Together, they shape the Tota-Bijagual-Mampacha paramo complex [27]. Precipitation values range between 1000 and 2500 mm/year, and temperatures range between 13 and 17 °C [37].

### 4.2. Data Source

Satellite data interpreted with landscape metrics (i.e., diversity, composition, and configuration) were used to analyze the spatial and temporal dynamics of the Bijagual Massif vegetation. The vegetation cover thematic maps were edited according to the method proposed by The Nature Conservancy [71] and Posada [44] at a scale of 1:100,000. The satellite-derived data was obtained from three multispectral Landsat images (available in: Earth Explorer of the USGS—U.S. Geological Survey): 1. Landsat 5 Thematic Mapper (TM; LT50070561986013; year 1986); 2. Landsat 7 Enhanced Thematic Mapper (ETM+; LT700705620001213; year 2000), and 3. Landsat 8 Operational Land Imager (OLI; LC08L2SP00705620211113-25; year 2021). These images visually represent the information captured by each sensor (vegetation covers in this research) across different regions of the electromagnetic spectrum [22,25,26]. They have spatial (pixel size: 30 m), spectral (number of bands), radiometric (Digital Levels: DL), and temporal (available every 16 days) resolutions that allowed for the comparison of the covers (Table 3 [72]). The images are located in Path/Row 7/56 according to the Landsat grid and have cloudiness < 10% [22,25,26]. Band combinations in each image were made according to the specifications of each sensor to differentiate covers. For the 1986 and 2000 images, the combination was Red: 4, Green: 5, and Blue: 3, and for 2021, Red: 5, Green: 6, and Blue: 2. Pre- and post-processing was performed with the programs ERDAS Image ver. 2016 and ArcGIS ver. 10.7 (academic licenses) described below.

### 4.3. Geometric Correction

One hundred and fifty control points (in images 1986 and 2021) were generated from a third-order polynomial model and a georeferenced image (image from 2000; [22,44]); in the 1986 image, 21 automatic points and 129 manual points were projected (standard error-RMSE 0.59; Std. Dev: 0.218); in 2021, 10 automatic and 128 manual points (RMSE 0.55; Erros Std. Dev: 0.208) were projected. These were considered reliable because their values were <1.0 [44]. Subsequently, the study area was cut from the Bijagual polygon.

### 4.4. General Contrasts (Spatial Enhancement)

The images were smoothed in their spatial contrasts similar to the digital Level of each pixel with respect to neighboring pixels using low-pass filters (Kernel 3 × 3) to reduce spatial variability and visualize less sharp, more blurred, and homogeneous profiles.

### 4.5. Land Covers (Thematic Categorization)—Field Verification

A statistical digital classification was performed for each image to group the pixels by cover, i.e., going from a continuous measurement (digital levels of each pixel) scale to a categorical one (vegetation coverage in the Corine Land Cover system—IDEAM [33]) using the supervised classification method and the selection and delimitation of training areas (grouping of pixels of the same category).

The supervised classification method was supported with field verification activities, which involved obtaining geographic coordinates (control points in the WGS84 system) and floristic and vegetation characterizations. During the years 2012–2016, transects were conducted in each natural cover type (27 of 250 m^2^ in Hof and 13 of 50 m^2^ in Dgnm each), and exhaustive surveys were carried out across the massif. Botanical samples were collected in each sampling unit (a total of 1781 exsiccates), and physiognomic data were compiled according to the method of Rangel and Velásquez [73] to describe the composition, structure, and biodiversity potential in Bijagual (published in Gil-Leguizamón et al. [34]). The botanical material was processed and deposited in the UPTC Herbarium, determined using specialized literature such as Gentry [74], Flora de Colombia, Flora Neotrópica, Flora de Costa Rica, Flora de Panamá, and corroborated with websites (www.tropicos.org, www.botanicus.org, and www.jstor.org, accessed between the years 2013–2021), as well as collections from the UPTC, COL, HECASA Herbaria, and support from specialists in the various groups. Additionally, for endemic and threatened species, Herbarium collections and online catalogs were consulted (published in Gil-Leguizamón et al. [35] and Carrillo et al. [37]).

### 4.6. Classification Accuracy

The reliability of the training areas (spectral separability of the categories) was statistically evaluated through an error matrix that contrasted the real classes (the truth of the terrain) and the cartographic units (cover). Additionally, accuracy measurements were used for each cover to determine the reliability of the supervised classification, including:

#### 4.6.1. Producer Accuracy

The omission error measure indicates the probability that a sample point is correctly classified. It is calculated by dividing the number of pixels correctly classified into a category by the total number of pixels in that category according to Equation (1).
(1)PP%=(Xii/X+i)×100
where ***PP*%** is producer accuracy in **%**, ***Xii*** is the value of the column diagonal, and ***X*** + ***i*** is the total marginals of column ***i***.

#### 4.6.2. User Accuracy

The probability that a sampling point corresponds to the category assigned in the field (commission error). It is calculated by dividing the total number of correct pixels in a category by the total number of pixels that, in fact, belong to this category according to Equation (2).
(2)PU%=(Xii/Xi+)×100
where ***PU*%** is user accuracy in **%**, ***Xii*** is the diagonal value of row ***I***, and ***Xi*****+** is the total marginals of this row.

#### 4.6.3. Kappa Coefficient

It measures the difference between what is observed between classes and what is evidenced in the classification according to Equation (3). The veracity corresponds to values close to or equal to 1.
(3)K=(O−E)/(1−E)
where ***O*** is the correct proportion of the classes identified in the image, and ***E*** is the correct observed percent change estimate.

### 4.7. Multitemporal Dimension (Landscape Metrics)

Statistics to identify the spatial and temporal dynamics of the coverages for 1986, 2000, and 2021 were obtained according to the spatial pattern theory [75]. This incorporates the landscape as a mosaic of patches or discrete covers, which explain the distribution of geographic objects, their patterns, and processes in time [75,76]. The statistics used quantified the landscape with three parameters: diversity and composition with Fragstats [75] and connectivity with Conefor [66,67,77], as follows:

#### 4.7.1. Diversity

It was analyzed through the total landscape area (TA, ha), the total number of patches (TNP), patch richness (PR), and patch richness density (PRD).

#### 4.7.2. Composition

Includes the area of each cover or class (CA, ha), the number of patches (NP), patch density (PD), landscape percentage (PLAND, %), and the largest patch index (LPI, %) to analyze the composition.

#### 4.7.3. Connectivity

The topo-ecological index of connectivity probability (***dPC_k_***) was used and interpreted according to three fractions that quantified how the patches (as habitat or corridor) contribute to the global connectivity of the Bijagual landscape, according to Equation (4), to obtain connectivity information.
(4)$${dPC}_{k}={dPCintra}_{k}+{dPCflux}_{k}+{dPCconnector}_{k}$$
where ***k*** is a landscape element (patch), ***intra*** is the contribution of patch ***k*** in terms of the habitat area (or quality) available within it (***intrapatch connectivity***), and ***flux*** corresponds to the dispersion flux (weighted by the area) received or originated through the connections of patch ***k*** with the rest of the patches (it reflects how well-connected ***k*** is with the rest of the existing habitat in the landscape). ***Connector*** evaluates the contribution of ***k*** as a connecting element or bridge patch between the rest of the patches (estimates to what extent ***k*** facilitates dispersive flows and which do not have their origin or destination in ***k*** but are enhanced and pass through ***k***) [67,78]. Furthermore, an intermediate dispersal distance (0.5 probability) and an approximate average distance between patches of 50 m were assumed. The results are shown as bar and scatter plots obtained with the ggplot2 package [79] included in the open-source RStudio Desktop version software of 2003 for Windows 10/11.

## Figures and Tables

**Figure 1 plants-13-00948-f001:**
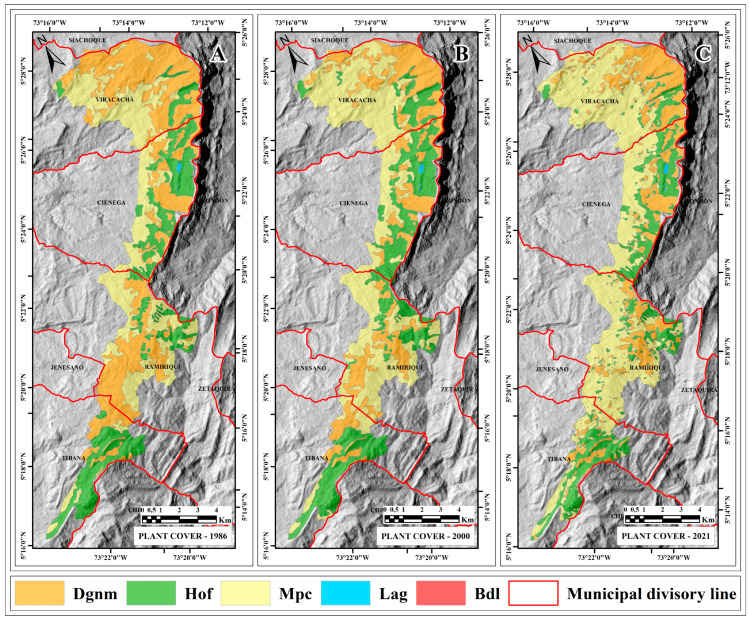
Composition and spatial configuration of the vegetation cover in the Bijagual Massif; (**A**): Cover in 1986, (**B**): Cover in 2000, and (**C**): Cover in 2021; Dgnm: Dense grasslands of non-wooded mainland (paramo); Hof: High open forest (high Andean forest); Mpc: Mosaic of pastures and crops; Lag: Lagoons; Bdl: Bare and degraded lands (deforested areas).

**Figure 2 plants-13-00948-f002:**
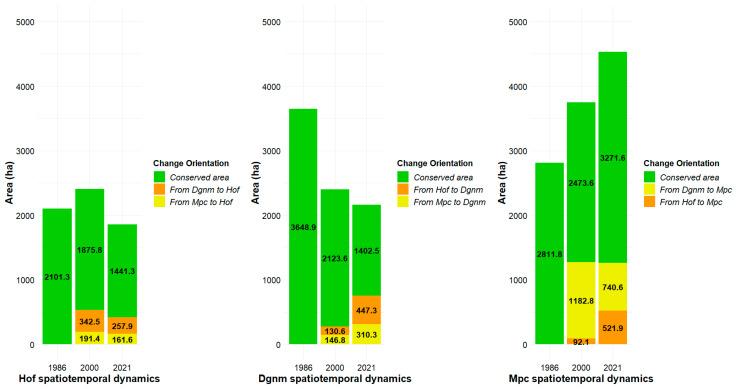
Spatiotemporal dynamics by land cover in the Bijagual Massif. Hof: High open forest; Dgnm: Dense grassland of non-wooded mainland; Mpc: Mosaic of pastures and crops; Lag areas (Lagoons) and Bdl (Bare and degraded lands) are not included in the graphs due to their low representativeness in terms of area.

**Figure 3 plants-13-00948-f003:**
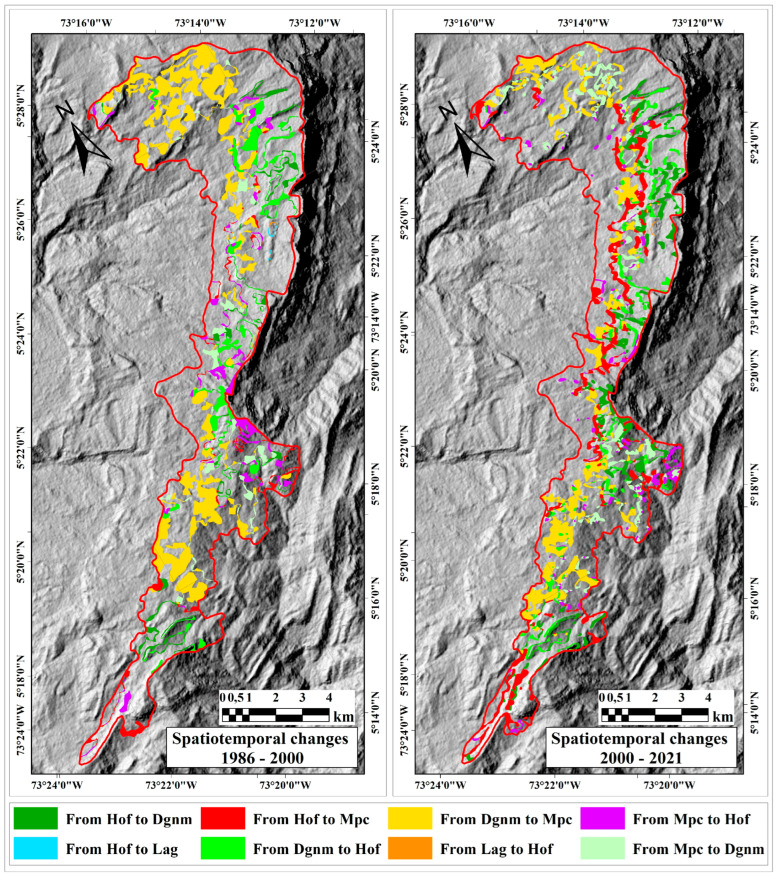
Orientation of spatiotemporal changes among land cover types for the Bijagual Massif.

**Figure 4 plants-13-00948-f004:**
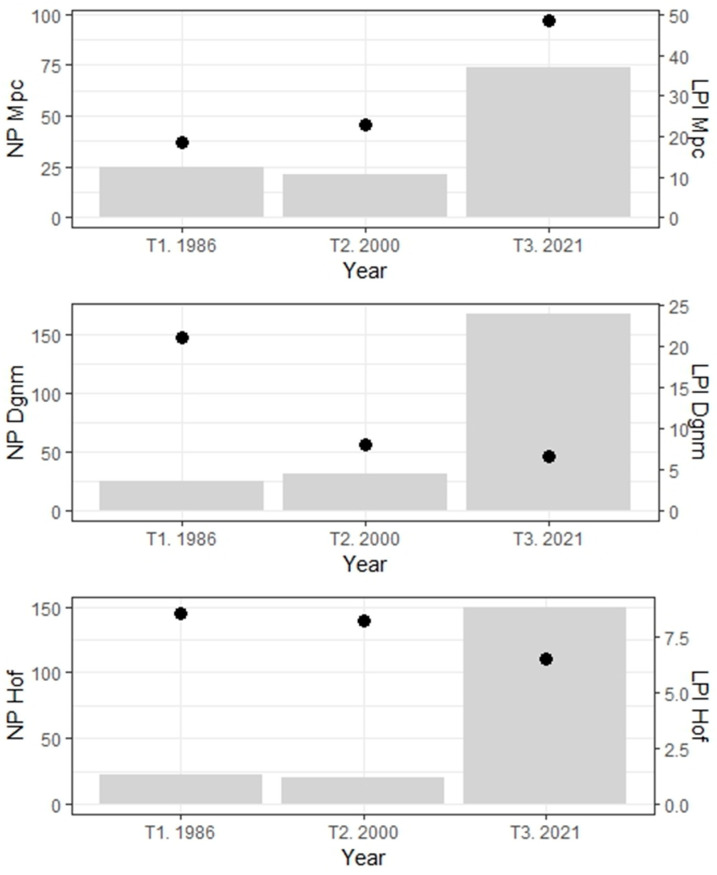
Temporal trend between the number of patches (NP in gray bars) and the largest patch index (LPI in black dots) for the mosaic of pastures and crops (Mpc), dense grassland of non-wooded mainland, i.e., paramo (Dgnm), and high Andean forest (Hof). T1-3: Temporality (years 1986, 2000, and 2021).

**Figure 5 plants-13-00948-f005:**
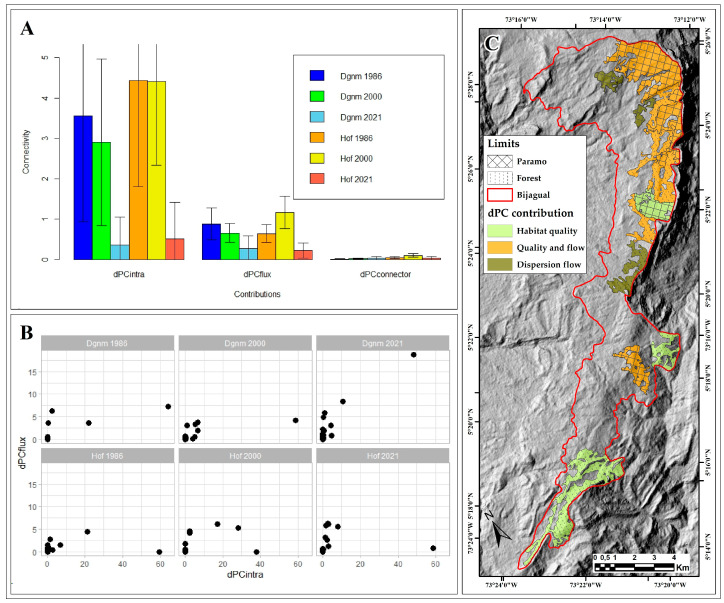
Ecological connectivity (dPC) in the Bijagual landscape. (**A**). Average contribution and standard deviation of the *dPCintra*, *dPCflux*, and *dPCconnector* fractions by cover and year; (**B**). Spatiotemporal contribution of the Hof and Dgnm patches to habitat quality and availability (*dPCintra*) and ecological flow (*dPCflux*); (**C**). Hof and Dgnm patches with the highest contribution to ecological connectivity by 2021. Dgnm: Dense grassland of non-wooded mainland (paramo); Hof: High open forest (high Andean forest).

**Figure 6 plants-13-00948-f006:**
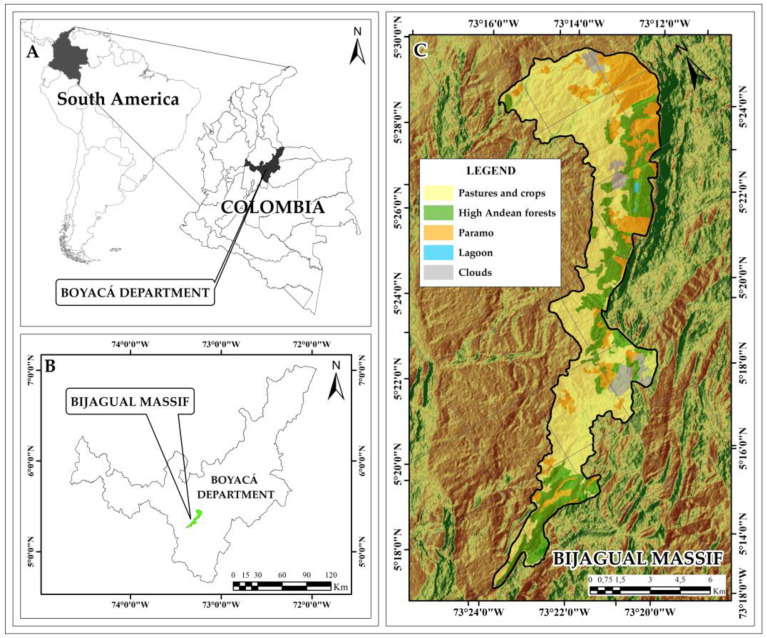
Study area. Geographic location of the Bijagual Massif in Colombia. (**A**). Location of the Boyacá department in Colombia and South America; (**B**). Location of the Bijagual Massif in the Boyacá Department; (**C**). Bijagual Massif.

**Table 1 plants-13-00948-t001:** Error matrix, thematic accuracy for 2021. The gray cells are correct assignments; the rest are leak assignments. Dgnm: Dense grassland of non-wooded mainland; Hof: High open forest; Mpc: Mosaic of pastures and crops; Lag: Lagoons; Bdl: Bare and degraded lands.

Classification	Dgnm	Hof	Mpc	Lag	Bdl	Total Row	User Accuracy (%)	Commission Error (%)
**Dgnm**	49	4	0	0	0	53	98.55	1.45
**Hof**	1	140	0	2	0	143	98.91	1.09
**Mpc**	0	2	71	0	0	73	99.28	0.72
**Lag**	0	0	0	6	0	6	100	0
**Bdl**	0	0	0	0	1	1	100	0
**Total column**	50	146	71	8	1	276		
**Producer accuracy (%)**	99.64	97.83	100	99.28	100	Overall classification accuracy (Kappa) = 0.95		
**Error of omission (%)**	0.36	2.17	0	0.72	0			

The residuals in the paramo, forest, pasture, and lagoon columns indicate real coverage that is not included in the map (error of omission, when a pixel has a specific coverage on the terrain and is not assigned to that class on the map); the residuals of the rows were associated with map coverages that do not conform to reality (error of commission, elements that, although they do not belong to a particular class, are found in it).

**Table 2 plants-13-00948-t002:** Area occupied per vegetation cover for the years 1986, 2000, and 2021 in Bijagual. Mpc: Mosaic of pastures and crops; Dgnm: Dense grassland of non-wooded mainland; Hof: High open forest; Lag: Lagoons; Bdl: Bare and degraded lands.

Cover	Area (ha)						Change from 1986 to 2000 (ha)	Change from 2000 to 2021 (ha)	Change from 1986 to 2021 (ha)
	1986	%	2000	%	2021	%			
**Mpc**	2811.8	32.8	3748.5	43.7	4534.1	52.9	936.6	785.6	1722.2
**Dgnm**	3648.9	42.6	2400.9	28.0	2160.0	25.2	−1248.0	−241.0	−1489.0
**Hof**	2101.3	24.5	2410.5	28.1	1861.8	21.7	309.2	−548.7	−239.5
**Lag**	12.1	0.1	14.1	0.2	13.1	0.2	2.0	−1.0	1.0
**Bdl**	0.0	0.0	0.0	0.0	5.1	0.1	0.0	0.9	0.9
**Total**	8574.1	100.0	8574.1	100.0	8574.1	100.0			

**Table 3 plants-13-00948-t003:** Multispectral characteristics of processed Landsat images. The spectral ranges (µm) correspond to the combined bands.

Satellite Sensor (Image)	Resolution		
	Spatial	Spectral	Radiometric
Landsat 5 TM (LT50070561986013)	30 m (visible, NIR, SWIR) 120 m (thermal)	Number of bands: 7	8 bits, 28: 256 DL
		Band 3: 0.63–0.69 µm	
		Band 4: 0.76–0.90 µm	
		Band 5: 1.55–1.75 µm	
Landsat 7 ETM+ (LT700705620001213)	30 m (visible, NIR, SWIR) 60 m (thermal) 15 m (panchromatic)	Number of bands: 8	8 bits, 28: 256 DL
		Band 3: 0.63–0.69 µm	
		Band 4: 0.775–0.90 µm	
		Band 5: 1.55–1.75 µm	
Landsat 8 OLI (LC08L2SP00705620211113-25)	30 m (visible, NIR, SWIR) 100 m (thermal) 15 m (panchromatic).	Number of bands: 11	16 bits, 216: 65,536 DL
		Band 2: 0.452–0.512 µm	
		Band 5: 0.851–0.879 µm	
		Band 6: 1.566–1.651 µm	

## Data Availability

All data supporting the reported results are included in the paper. Biological records registered in each coverage are available in SibColombia: https://ipt.biodiversidad.co/cr-sib/pdf.do?r=0724_bijagual_20180824&n=1656C6909E7.

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
