# Peer review of "Spatial and Temporal Dynamics of the Vegetation Cover from the Bijagual Massif, Boyacá, Colombia, during the 1986–2021 Period"

_plants, 2024, doi:10.3390/plants13070948_

Round 1

Reviewer 1 Report

Comments and Suggestions for Authors

The paper presented an interesting study on how to analyze the spatio-temporal dynamics of the vegetation cover in Bijagual Massif. There is interesting work in this article. I’m still confused about some issues.

1. Please consider the adequacy of the keywords of the article, e.g., the research topic of the article: land use change.

2. Information needs to be supplemented. For example, ‘Between 1986 and 2000, Dgnm lost 1,248 ha (34%), while Mpc and Hof increased 936.6 ha (25%) and 309.2 ha (13%), respectively.’ (Line 144-145) data marked in blue need to be supplemented.

3. The authors can describe the transition matrix of Table 3 in pictures.

4. Please harmonize the format of the contents of the brackets (Line 188-196). For example, (74/PD:0.86) to (NP: 74 & PD: 0.86 patches/ha).

5. Please describe in detail the timing and methodology of the fieldwork.

6. Please reconsider whether the three years (1986, 2000, 2021) of land use classification data used in the current study are sufficient for the study, as many land use types change multiple times in a short period (e.g., deforestation leading to a change in land type to wasteland, and then to scrubland or forests).

If authors can solve the above problems well, I think this work can be reconsidered.

Author Response

Observation

Reply

1. Please consider the adequacy of the keywords of the article, e.g., the research topic of the article: land use change.

The observation is accepted: the keyword "Land Use Change" is included (line 32).

2. Information needs to be supplemented. For example, ‘Between 1986 and 2000, Dgnm lost 1,248 ha (34%), while Mpc and Hof increased 936.6 ha (25%) and 309.2 ha (13%), respectively.’ (Line 144-145) data marked in blue need to be supplemented.

The observation is accepted: adjustments are made in the wording. The article specifies that the percentage was obtained from the total estimated area for the Bijagual Massif (lines 159 and 160).

3. The authors can describe the transition matrix of Table 3 in pictures.

The observation is accepted: Table 3 is replaced by Figure 2. This figure describes the spatiotemporal dynamics (area changes) for each land cover. In this figure, the green bars depict areas that remain over time, while the yellow and brown colours represent gained areas, i.e., areas that one land cover yields to another (lines 206-210). Wording adjustments are made (lines 177-204).

4. Please harmonize the format of the contents of the brackets (Line 188-196). For example, (74/PD:0.86) to (NP: 74 & PD: 0.86 patches/ha).

The observation is accepted: wording adjustments are made, and abbreviations for metrics are included in parentheses (lines 225-235).

5. Please describe in detail the timing and methodology of the fieldwork.

The observation is accepted: in the methodology, the information is expanded, and field activities are detailed in terms of sampling units, herbarium curation, and associated publications supporting field verification (lines 502-516).

6. Please reconsider whether the three years (1986, 2000, 2021) of land use classification data used in the current study are sufficient for the study, as many land use types change multiple times in a short period (e.g., deforestation leading to a change in land type to wasteland, and then to scrubland or forests).

The observation is not accepted: we agree with the reviewer's comment; indeed, it is possible to identify more change orientations for other years and with other satellite images not included in this research. However, the landscape composition of Bijagual (the type and number of land covers) did not change drastically during the evaluated periods; the four main covers (Hof, Dgnm, Mpc, and Lag) have remained for 35 years, being the subject of land use dynamics in the natural area. Only in 2021 does an exposed soil cover (Bdl: Bare and degraded lands) appear, with an area (5.1 ha) that is not significant at a 1:100000 scale to describe representative changes in the landscape configuration analyzed (orientation of changes). Therefore, the three processed images were sufficient to describe the influence of anthropogenic activities and the deterioration of natural cover conditions (evaluated with landscape metrics).

Andean high-mountain ecosystems present relatively low resilience; a cover change has a slow recovery impact (Prado-Castillo et al., 2018). The satellite images for the evaluated periods (1986-2000, 2000-2021) allowed us to obtain information that, at a landscape scale, allowed us to assume recovery processes, loss or gain of area, and possible ecological successions (the latter may be the subject of other research at a 1:1 scale). Similarly, the article's approach seeks to raise awareness of how intervention without planning leads to the deterioration of natural areas, which provide ecosystem services and are not easily recovered.

The processed satellite images were inputs to interpret the transformation of the territory; however, a field verification process (1:1 scale) is included as support for the current reality of the area (accuracy of information) and provides detail to the obtained thematic maps. Although the geographical scale worked on is general (1:100000), the change was detailed with vegetation information obtained at a 1:1 scale and verified in the field and laboratory, which supported the analysis of the changes.

Reviewer 2 Report

Comments and Suggestions for Authors

Based on Landsat data, this study analyzes the spatiotemporal variation characteristics of vegetation in Bijagual Massif from 1986 to 2021. The research results may contribute to further understand the driving mechanisms of spatiotemporal changes in vegetation in Bijagual Massif. However, there are some concerns that the authors should address before it can be considered for publication.

1. Lines 42-44, I suggest the authors add the analysis of the impact of climate change on primitive forest landscapes.

2. In the last paragraph of the introduction, I suggest the authors highlight the significance of this study.

3. In the data, I suggest the authors add more information about data, such as  Landsat data availability and access.

4. In section 2.3, the authors should add a spatial distribution map of vegetation change from 1986 to 2021.

5. More mechanism explanations are encouraged to further explain the relationship between the vegetation cover and climate change.

6. This study lacks exploration of the underlying reasons behind the spatiotemporal changes of vegetation in Bijagual Massif. Therefore, I suggest that the author strengthen their research on the driving mechanisms of spatiotemporal vegetation changes in Bijagual Massif.

7. In the uncertainty, I suggest the authors further discuss the uncertainty of remote sensing data including NDVI data (e.g., Wang et al., 2021; Ma et al., 2022) which may affect the research results.

References:

Spatiotemporal change of aboveground biomass and its response to climate change in marshes of the Tibetan Plateau. International Journal of Applied Earth Observation and Geoinformation, 2021, 102: 102385.

Variation of vegetation autumn phenology and its climatic drivers in temperate grasslands of China. International Journal of Applied Earth Observation and Geoinformation, 2022, 114: 103064.

Author Response

Observation

Reply

1. Lines 42-44, I suggest the authors add the analysis of the impact of climate change on primitive forest landscapes.

The observation is accepted: a paragraph describing the influence of climate change on natural covers is included in the introduction (lines 46-55):

“To these directly caused human transformations, other indirectly caused ones due to the current situation of global climate change are added, mainly, the constant increase in the average annual temperature and the decrease in precipitation over the last decades. These climatic anomalies cause, among other effects, alterations in the natural dynamics of vegetation, changes in the structure and composition of their communities, and a notable loss of biodiversity (Gao et al., 2016; Ometto et al., 2022; Reyes-Palomino et al., 2022). In the case of paramo formations, given their high sen-si-tivity to environmental disturbances, it is estimated that more than three-quarters of the original ecosystem has been altered due to the increase in temperatures, which dries the soil and forces species to move along the altitudinal axis towards areas more suitable for their development (López E. I., 2012; Añarumba & Toapanta, 2023)”.

2. In the last paragraph of the introduction, I suggest the authors highlight the significance of this study.

The observation is accepted: the importance of the study conducted is included in the introduction (lines 74-84).

“The objective of the above analysis is to provide reference values for the conservation, ecology, sustainability, vegetation management, and territorial planning of the last remnants of forest and paramo existing in the Bijagual Massif, which is considered an ecological corridor, regulating high-mountain water and climate in Colombia [4,5,9,11,22,25]. Based on remote sensing data, we explore the supervised classification method in two periods spanning 35 years, from 1986-2000 and 2000-2021. This study is based on: 1. Describe the natural high-mountain vegetation covers in Bijagual as an indicator of their potential biodiversity. 2. Validate the thematic reliability (spectral separability) of the current vegetation covers. 3. Identify spatiotemporal changes (losses and gains) in high-mountain natural areas and their influence on ecological connectivity”.  

3. In the data, I suggest the authors add more information about data, such as Landsat data availability and access.

The observation is accepted: in the Materials and Methods section, section 4.2 Data source, information related to the availability of Landsat data is expanded, and the characteristics of spatial, temporal, spectral, and radiometric resolution of each satellite image used in the research are listed (a table is included; lines 459-477).

4. In section 2.3, the authors should add a spatial distribution map of vegetation change from 1986 to 2021.

The observation is accepted: a figure showing the orientation of changes between land covers for the periods 1986-2000 and 2000-2021 is included (Figure 3; lines 212-217).

5. More mechanism explanations are encouraged to further explain the relationship between the vegetation cover and climate change.

The observation is accepted: in the Discussion section, the information supporting the results of this research is expanded according to the patterns of vegetation distribution influenced by climate change. Bibliographic references are included (lines 358-379).

6. This study lacks exploration of the underlying reasons behind the spatiotemporal changes of vegetation in Bijagual Massif. Therefore, I suggest that the author strengthen their research on the driving mechanisms of spatiotemporal vegetation changes in Bijagual Massif.

The observation is not accepted: the observation requested by the reviewer is already included in the discussion, where mechanisms such as:

The land use history in the Bijagual Massif is primarily focused on agricultural and livestock activities; however, the extent of changes in natural areas resulting from these activities is unknown. In Colombian high-mountain ecosystems (in the departments of Boyacá, Cundinamarca, and Santander), potato cultivation has been a tradition for farmer peasants, and they engage in this agricultural practice as a means of livelihood. Given this context, the discussion in the document guides the reader to identify the extent of change resulting from excessive and unplanned soil use.

Furthermore, we describe how intensive and extensive soil change limits natural area recovery, leading to fragmentation processes evidenced by the distancing of natural areas, loss of area, and decreased ecological connectivity in the biological corridor. We include information linking climate change as a transformative phenomenon in natural landscapes, which has led to vegetation migration processes and the need for ecological restoration strategies (with research, action, and participation from different social sectors).

Based on these arguments, we consider including the mechanisms that have driven changes in land covers in Bijagual.

7. In the uncertainty, I suggest the authors further discuss the uncertainty of remote sensing data including NDVI data (e.g., Wang et al., 2021; Ma et al., 2022) which may affect the research results.

References:

Spatiotemporal change of aboveground biomass and its response to climate change in marshes of the Tibetan Plateau. International Journal of Applied Earth Observation and Geoinformation, 2021, 102: 102385.

Variation of vegetation autumn phenology and its climatic drivers in temperate grasslands of China. International Journal of Applied Earth Observation and Geoinformation, 2022, 114: 103064.

The observation is not accepted: the Normalized Difference Vegetation Index (NDVI) is used to estimate the quantity and health of vegetation (MappingGIS, 2020; Pettorelli et al., 2005; Chuvieco, 2002; Rouse, 1974). It is sensitive to the presence of chlorophyll, making it particularly useful for identifying areas with high vegetation density. However, in areas with very high vegetation density (such as the study area), NDVI can become saturated, limiting its ability to discriminate between different types of vegetation (Testa et al., 2018).

This index can be useful for differentiating natural covers from those intervened by humans; however, for the study area (due to spatial heterogeneity), it does not differentiate between natural covers, i.e., spectral differentiation between forest (Hof) and páramo (Dgnm). This index would limit the analysis of changes in natural areas; instead, the supervised classification method (Posada, 2012) allowed for the identification of spectral and spatial differences between all covers. Therefore, uncertainty was assumed with the error matrix (Table 1), which contrasted spectral separability with field verification (omission and commission errors; Lines 166-172) plus the estimation and interpretation of the Kappa statistic (Lines 146-149; 312-326).

References:

Chuvieco, E. Teledetección ambiental, la observación de la Tierra desde el espacio. Editorial Ariel S.A.: Barcelona, Spain, 2002; pp. 586.

MappingGIS. (2020). Los 6 índices de Vegetación para completar el NDVI - MappingGIS.

https://mappinggis.com/2020/07/los-6-indices-de-vegetacion-para-completar-el-ndvi/

Pettorelli, N., Vik, J. O., Mysterud, A., Gaillard, J.-M., Tucker, C. J., & Stenseth, N. Chr. (2005). Using the satellite-derived NDVI to assess ecological responses to environmental change. Trends in Ecology & amp; Evolution, 20(9), 503-510.

Posada, E. Manual de prácticas de percepción remota parte 2. Centro de Investigación y Desarrollo en Información Geográfica (CIAF). Instituto Geográfico Agustín Codazzi (IGAC): Bogotá, Colombia, 2012; pp. 154.

Rouse, J.W., et al. Monitoring Vegetation Systems in the Great Plains with ERTS. NASA Special Publication, 1974; pp. 351.

Testa, S., Soudani, K., Boschetti, L., & Borgogno Mondino, E. MODIS-derived EVI, NDVI and WDRVI time series to estimate phenological metrics in French deciduous forests. International Journal of Applied Earth Observation and Geoinformation. 2018, 64, 132-144.

Reviewer 3 Report

Comments and Suggestions for Authors

This research delves into the transformative shifts within the highly endemic Andean forest and Paramo ecosystem in Bijagual Massif (Colombia) over a span of 35 years. Employing an in-depth examination of spectral imagery data from satellite photographs, the study meticulously constructs diachronic vegetation maps covering an expansive 8,574.1 hectares. Furthermore, it conducts a comparative analysis of crucial ecosystem functioning parameters, including the diversity and composition of plant communities and habitat connectivity.

The findings reveal a substantial decline in natural habitats such as forests and paramo, giving way to the expansion of pastures and cultivation, accompanied by the emergence of severely degraded areas devoid of any plant cover. Ecosystem indicators underscore a heightened fragmentation of habitats, rendering the resident populations increasingly vulnerable. The article sheds light on the dynamics of vegetation in previously cultivated regions, addressing the deforestation linked to agricultural development and its correlation with local farming practices.

Additionally, the study discusses the ramifications of altered habitat connectivity and its potential implications for the ecological balance of these ecosystems. An integral aspect of this study would involve examining the establishment of local herbivore populations in response to the evolving landscape, providing a comprehensive perspective on the ecosystem's overall health and sustainability.

Comments :

Figure 1. Composition and spatial configuration of the vegetation cover in the Bijagual Massif : “Bdl: Bare and degraded lands” are mentioned as caption but not visible on the maps. Maybe is due to the resolution of the figure since this vegetation cover area is very low as explained by the authors further (see table 2).

 Figure 3. Ecological connectivity in the Bijagual landscape: indices of dPC contribution should be detailed in the caption.

Line 187 “Composition:” should be replaced by “Composition” without “:”

Author Response

Observation

Reply

Figure 1. Composition and spatial configuration of the vegetation cover in the Bijagual Massif: “Bdl: Bare and degraded lands” are mentioned as caption but not visible on the maps. Maybe is due to the resolution of the figure since this vegetation cover area is very low as explained by the authors further (see table 2).

The observation is accepted: we agree with the reviewer's comment; indeed, due to the resolution of Figure 2 and the estimated area for the "Bdl: Bare and degraded lands" coverage (5.1 ha), it cannot be visualized in the figure. However, Table 2 includes the area of the coverage and the year in which it was identified.

Figure 3. Ecological connectivity in the Bijagual landscape: indices of dPC contribution should be detailed in the caption.

The observation is accepted: an adjustment is made to the title of Figure 5, including the acronym for ecological connectivity (dPC; line 269):

"Figure 5. Ecological connectivity (dPC) in the Bijagual landscape…"

The specification of the fractions composing the dPC index is described in detail in the methodology.

Line 187 “Composition:” should be replaced by “Composition” without “:”

The observation is accepted: an adjustment is made to the document, removing ":".

Round 2

Reviewer 2 Report

Comments and Suggestions for Authors

The authors have addressed all my concerns. I suggest accept this paper in its present form.